# Are Higher Body Mass Index and Worse Metabolic Parameters Associated with More Aggressive Differentiated Thyroid Cancer? A Retrospective Cohort Study

**DOI:** 10.3390/healthcare12050581

**Published:** 2024-03-02

**Authors:** Yasmin Abu Arar, Michael Shilo, Natalya Bilenko, Michael Friger, Hagit Marsha, David Fisher, Merav Fraenkel, Uri Yoel

**Affiliations:** 1Internal Medicine Ward D, Soroka University Medical Center, Beer-Sheva 84101, Israel; yasminab@tlvmc.gov.il; 2Department of Epidemiology, Biostatistics and Community Health, Ben-Gurion University of the Negev, Beer-Sheva 84015, Israel; shilomic@post.bgu.ac.il (M.S.); natalya@bgu.ac.il (N.B.); friger@bgu.ac.il (M.F.); 3Faculty of Health Sciences, Goldman Medical School, Ben-Gurion University of the Negev, Beer-Sheva 84015, Israel; marsha@post.bgu.ac.il; 4Endocrinology Unit, Soroka University Medical Center, Beer-Sheva 84101, Israelmeravfre@clalit.org.il (M.F.); 5Faculty of Health Sciences, Ben-Gurion University of the Negev, Beer-Sheva 84015, Israel

**Keywords:** differentiated thyroid cancer (DTC), papillary thyroid carcinoma (PTC), PTC histological variants, obesity, body mass index (BMI), metabolic syndrome, triglycerides/high-density lipoprotein (HDL)–cholesterol ratio

## Abstract

Obesity is a risk factor for differentiated thyroid cancer (DTC), but the association with DTC aggressiveness is controversial. To evaluate the association between preoperative body mass index (BMI)/other metabolic parameters and DTC aggressiveness in our surgical cohort, we retrospectively evaluated patients following thyroid surgery who were diagnosed with DTC between December 2013 and January 2021. Baseline characteristics, histopathological features, treatment modalities, and follow-up data were studied. We conducted logistic regression to analyze the association between BMI/other metabolic parameters and adverse DTC features. The final study cohort included 211 patients (79.6% women; mean age± standard deviation 48.7 ± 15.9 years): 66 (31.3%) with normal weight, 81 (38.4%) with overweight, and 64 (30.3%) with obesity. The median follow-up was 51 months (range 7–93). Complete versus partial thyroidectomy was more common among patients living with overweight or obesity than in normal weight patients (79.7% versus 61.7%, *p* = 0.017, respectively). Logistic regression demonstrated that higher BMI was associated with mildly increased risk for lymph nodes metastases (odds ratio [OR] 1.077, 95% CI: 1.013–1.145), and higher triglycerides/high-density lipoprotein–cholesterol (TG/HDL-C) ratio was associated with aggressive histological variants of DTC (OR 1.269, 95% CI 1.001–1.61). To conclude, specific adverse clinical and histopathological DTC features were indeed associated with higher BMI and higher TG/HDL-C ratio.

## 1. Introduction

Obesity is anthropometrically defined as a body mass index (BMI) ≥ 30. The global burden of obesity has more than tripled since 1975 and is continuing to increase around the world [1,2]. Cardiovascular and metabolic diseases are the leading causes for morbidity and mortality among individuals with overweight (BMI 25–30) and obesity [3,4,5]. In addition to cardiometabolic complications, it has been demonstrated that the prevalence of obesity-related cancer is also increasing, with evidence showing that excess body weight and metabolic syndrome are associated with an increased risk of breast, gastrointestinal, gynecological, and thyroid cancers (TC) [6,7]. Insulin resistance and adipocytokines that are secreted from unhealthy adipose tissue appear to have a key role in cancer pathogenesis [7]. Recently, it was demonstrated that women with obesity with early breast cancer were more likely to have genomic alterations which were associated with inflammation and aging of the breast cancer tumor microenvironment [8].

TC is the most common endocrine malignancy which encompasses a variety of histotypes that are heterogeneous in frequency and clinical behavior. Differentiated thyroid cancer (DTC) refers to thyroid neoplasms derived from follicular cells, including papillary, follicular, and Hürthle cell TCs (PTC, FTC, and HCC, respectively). PTC is the most common subtype, accounting for approximately 85% of new TC cases, and has the best prognosis compared with other TCs, with a 10-year survival exceeding 90%. Rarer histotypes include aggressive PTC variants such as tall cell and columnar cell, and moderately and poorly differentiated TC. These histotypes are more aggressive and carry a worse prognosis [9,10]. Established risk factors for DTC include female gender, family history of DTC, personal history of irradiation exposure [11] and increased body weight [7,12]. Partial or complete thyroidectomy is usually the first-line treatment for patients diagnosed with DTC ≥ 1 cm, followed by radioactive iodine (RAI) treatment, when indicated. Rarely, external irradiation is used to treat an unresectable cancerous mass. Recently, targeted therapy using kinases inhibitors has become available for patients with recurrent, aggressive, non-iodine-avid TC [11,13].

Epidemiologically, there is robust evidence for an association between overweight, obesity and metabolic syndrome and the risk of DTC [12,14,15,16,17,18,19]. The risk of TC has been found to be increased by up to 55% in individuals with overweight and obesity compared with people with normal weight [14]. Since stronger associations between obesity and TC mortality have been observed than those between obesity and TC incidence, detection bias is thought to be unlikely [20]. The link between obesity and worse metabolic parameters and DTC aggressiveness is controversial. Studies using a mouse model of a relatively aggressive form of FTC, the second most common histological sub-type of DTC, demonstrated that a high-fat diet and excess fat tissue were associated with more invasive FTC and a higher tendency to develop distant metastases. Intriguingly, these effects were (partially) blocked with metformin, suggesting that insulin resistance is causative, and by the inhibition of the signal transducer and activator of transcription 3 (STAT3) signaling pathway, which is largely activated by leptin [21,22,23]. In human studies, however, where most DTC cases are typically slow-growing and have a favorable prognosis [9,10,11], results regarding the association between obesity and DTC aggressiveness are conflicting. Matrone and his colleagues evaluated 1058 patients with DTC who were treated with total thyroidectomy and RAI and found no association between BMI and DTC aggressiveness [24]. Conversely, a systematic review and meta-analysis, which included 11 retrospective cohort studies comprising 26,196 participants, demonstrated that higher BMI was associated with larger tumors, multifocality, and extra-thyroidal extension (ETE) [25]. Interestingly, a case control study demonstrated that diabetes mellitus and insulin resistance were associated with more aggressive DTC characteristics (predominantly a higher rate of capsular invasion and lymph node (LN) metastases) [26].

Herein we present an in-depth analysis of our surgical cohort of patients with a postoperative histopathological diagnosis of DTC. We aimed to evaluate the association between BMI and other metabolic parameters (e.g., triglycerides/high-density lipoprotein-cholesterol (TG/HDL-C) ratio), which better represent insulin resistance status [27], and adverse DTC features indicated by histopathological results, the extent of surgery that was performed, the activity of radioactive iodine used, and follow-up data such as the response to treatment and the rate of disease recurrence. We hypothesized that higher BMI and worse metabolic parameters may demonstrate an association with adverse histopathological and clinical DTC features.

## 2. Material and Methods

### 2.1. Study Population

The current study was conducted in Soroka University Medical Center, a 1100-bed teaching, tertiary care referral hospital, located in Beer-Sheva, south Israel. The population treated in our institution is mainly of Caucasian origin with two major ethnic groups, Jewish and Arabs. We retrospectively evaluated consecutive patients aged 18 years and older, following thyroid surgery (partial or total thyroidectomy) who were operated on between December 2013 and January 2021 and had a final histology that reported DTC. Based on the World Health Organization standardized categories of BMI, patients were divided into 3 groups based on their pre-operative BMI: normal weight (18.5–24.9 kg/m^2^), overweight (25–29.9 kg/m^2^), and obese (≥30 kg/m^2^). Baseline characteristics were compared between groups with an emphasis on metabolic parameters. Histopathological characteristics which represent intermediate or high risk for recurrency according to the 2015 American Thyroid Association DTC guidelines [28] were evaluated and compared between groups. According to these guidelines, low risk features include intrathyroidal DTC of less than 4 cm in largest diameter without ETE, no lymphovascular invasion, no more than five small metastatic lymph nodes (LN) (<0.2 cm) and in the absence of extranodal extension or distal metastasis. Higher risk characteristics include tumors of 4 cm in largest diameter or greater, ETE, vascular invasion, more than five small LN or large LN with extranodal extension, and distal metastasis. In addition, the extent of surgery that was performed and the activity of radioactive iodine treatment used as well as follow-up data, including the response to treatment and the rate or disease recurrence, were compared between groups. Patients were excluded if the tumor diameter was less than 1 cm (most <1 cm tumors were incidentally found during an operation for presumed benign multinodular goiter), if the histopathological report was incomplete, and when there was less than 6 months of follow-up data following histopathological diagnosis (to facilitate evaluation of the efficacy of the primary treatment) (Figure 1). Patients without anthropometric data from the year preceding the thyroid surgery were also excluded.

### 2.2. Data Sources

Demographic, clinical (including baseline characteristics, the extent of surgery that was performed, the activity of radioactive iodine used, and follow-up data such as the response to treatment and the rate of disease recurrence), biochemical, and histopathological data were retrieved from our institutional electronic medical records. BMI was documented based on anthropometric data that were measured upon admission to the index surgery. When admission data regarding BMI were incomplete, community electronic data were accessed for the most recent measurements of weight and height.

### 2.3. Statistical Analysis

Quantitative variables are presented as mean ± standard deviation (SD), median and interquartile range (Q1–Q3), or frequency (percentage). We present comparisons between all three groups (normal weight, overweight, and obese) and, when relevant, also between two groups (normal weight and overweight, which comprised overweight and obese as a single group). For univariate analysis, statistical differences between the three BMI groups were calculated using ANOVA for normally distributed variables, as well as with Kruskal–Wallis test for non-normally distributed variables, followed by comparing each combination of two groups out of the three, using Bonferroni and Dunn’s tests accordingly post hoc. For comparing between two groups, Student’s T test was used for normally distributed variables as well as Mann–Whitney test for non-normally distributed variables. Differences in categorical variables were calculated using the Chi squared test as well as Fisher’s exact test. In addition, for all parameters, we repeated the analyses after restricting the cohort to patients between the ages of 30 and 70 years, which roughly represent the average age ± SD. We chose 30 years as the lower limit of this range, since age 30 years represents a point of increased prevalence of the metabolic syndrome (especially central obesity and hypertriglyceridemia) [29]. The upper limit of this range, 70 years, was chosen since we speculated that with more advanced age, other medical problems (e.g., cognitive decline and dementia [30]) might influence the metabolic status. Multivariate multinomial logistic regression was constructed for the three categorial dependent variables, as well as binary logistic regression for dichotomic variables in order to assess the association between metabolic indexes as continuous variables and histopathological features of the tumor, after controlling for age, sex, and ethnicity. A two-sided *p* value of 0.05 or lower was considered statistically significant. Analyses were performed using SPSS software ver. 26.0 (IBM, Armonk, NY, USA).

## 3. Results

### 3.1. Baseline Characteristics

A total of 211 patients (79.6% women; mean age± standard deviation 48.7 ± 15.9 years) were included in the final study cohort: 66 patients (31.3%) had a normal weight (BMI 18.5–24.9), 81 patients (38.4%) were overweight, and 64 patients (30.3%) were obese. Median follow-up was 51 months (range 7–93). Patients with a normal weight were younger than patients with overweight or obesity (mean± SD: 41.5 ± 15.4, 52.6 ± 16.3, 51.1 ± 13.5, respectively; *p* = 0.001 for the comparison between overweight versus normal weight, and *p* < 0.001 for the comparison between patients with obesity versus normal weight DTC). Compared with individuals with a normal weight, a diagnosis of hypertension was more common in patients with overweight and obese (6.1%, 37%, 29.7%, respectively; *p* < 0.001 for both, overweight and obesity versus normal weight), and the same was demonstrated for diagnoses of pre-diabetes and diabetes mellitus (9.1%, 13.6%, 26.6%, respectively; *p* = 0.007 for obesity versus normal weight, and *p* = 0.041 for obesity versus overweight). Metabolic parameters positively correlated with weight (median [Q1–Q3]): fasting glucose 5.05 mmol/L (4.77–5.47), 5.27 mmol/L (5–5.66), and 5.49 mmol/L (5.05–6.22), respectively, *p* < 0.001 (obese versus normal weight patients with DTC); TG/HDL-C ratio 2 (1.4–3.3), 2.4 (1.7–3.6) and 2.8 (2.1–5.6), respectively, *p* = 0.004 (obese versus normal weight patients with DTC). TSH level did not differ between the three BMI groups (Table 1).

### 3.2. Histopathological Features

Most patients with DTC in all three BMI groups demonstrated low-risk histopathological features. The absolute number of patients who had intermediate or high-risk histopathological features was low. For most histopathological features, a statistically significant difference between the three BMI groups could not be demonstrated (Table 2). However, a higher prevalence of aggressive histopathological features, which did not reach statistical significance, was observed with increasing weight. A higher rate of aggressive histological variants (mainly tall cell, columnar, and insular variants of PTC together with moderately and poorly differentiated variants) was observed with higher BMI (6.1%, 11.1% and 15.6%, respectively). The same was demonstrated for any ETE, any lymph node (LN) metastases, bilateral lateral neck LN metastases, and extra-nodal extension (the penetration of a metastasis outside of the LN capsule). However, gross ETE, which better reflects a higher risk of recurrence, was less common among patients with obesity, but the absolute numbers were small in all three groups and, therefore, underpowered to allow for comparison (Appendix A). Distal metastases were more common in patients with normal weight, a trend which was not statistically significant (Table 2).

### 3.3. Treatment Modalities and Follow-Up

Complete thyroidectomy was performed more often than partial thyroidectomy in individuals with overweight (81%) or obesity (73%) than in patients with a normal weight (65%) (*p* = 0.024 for the comparison between overweight versus normal weight) (Table 3). In subgroup analysis, complete thyroidectomy was performed significantly more often than partial thyroidectomy in patients aged 30–70 with overweight (comprising individuals with overweight and obesity) compared with those aged 30–70 with a normal weight (61.7% versus 79.7, *p* = 0.017) (Appendix A). There was no correlation between the rate of RAI treatment and the activity administered and BMI. A non-statistically significant trend was seen between the risk of disease recurrence and increasing BMI (for intermediate or high disease recurrence risk 37.9%, 48.1%, and 46.9%, for normal weight, overweight and obesity, respectively, *p* = 0.36) (Table 3), and for incomplete response to treatment (ages 30–70 years, 6.4% among normal weight versus 13.6% in patients with overweight and obese DTC, *p* = 0.11) (Appendix A). No patient in the normal BMI group underwent an additional intervention beyond the index thyroid operation and RAI treatment (when indicated), whereas a few patients in the higher BMI groups did (none, 5/6.2%, and 5/7.8%, respectively, *p* = 0.062 and *p* = 0.024 for overweight and obesity versus normal weight, respectively) (Table 3).

### 3.4. Logistic Regression Analyses

Multinomial logistic regression analysis, as well as logistic regression, was used to assess for an association between metabolic indexes (BMI, fasting glucose, TG, and TG/HDL-C ratio) and histopathological features of DTC, after controlling for age, sex, and ethnicity. Following adjustment for age, sex, and ethnicity, higher BMI was associated with a mildly increased risk of LN metastases (odds ratio [OR] 1.077, 95% CI:1.013–1.145) (Table 4). In patients aged between 30 and 70 years at the time of the index operation, higher TG/HDL-C ratio was associated with the more aggressive histological DTC variants (OR 1.269, 95% CI:1.001–1.61) (following adjustment for age, sex, and ethnicity) (Table 5). BMI, fasting glucose (Appendix A), TG levels (Appendix A) and TG/HDL-C ratio were not associated with any other histopathological variable. 

## 4. Discussion

The question of whether body weight and metabolic parameters are associated with DTC aggressiveness remains controversial. It was our aim to evaluate retrospectivity our cohort of patients with DTC to try and bridge this gap of knowledge. The rate of patients with DTC with adverse histopathological features was low in our cohort, a finding which is consistently demonstrated in cohorts of patients with DTC [24,31,32,33]. We observed a link, which did not achieve statistical significance, between higher BMI and several adverse histopathological features, including aggressive histological variants, any ETE, LN metastases, bilateral lateral neck LN metastases, and extra-nodal extension (Table 2). However, distal metastases occurred more commonly in patients with a normal weight, a trend which was not statistically significant. This observation might be the result of a catabolic state experienced by patients with advanced malignancy. More patients with BMI ≥ 25 underwent complete versus partial thyroidectomy, as compared with patients with DTC with a normal weight (79.7 versus 61.7%, *p* = 0.017), and were more likely to undergo an additional procedures. Logistic regression showed that a higher TG/HDL-C ratio was associated with aggressive histological variants (OR 1.269, 95% CI:1.001–1.61). Importantly, this association was statistically significant only when we analyzed patients aged 30–70 years, a range which roughly represents the average age of our cohort ±SD.

Several studies have addressed the association between higher BMI and adverse DTC features. Feng [34] evaluated more than 400 patients with PTC and found that higher BMI was associated with ETE (OR 6.1 [95% C.I. 1.8–20.8]) and vascular invasion (OR 3.9 [95% C.I. 1.1–14.3] and 9.2 [95% C.I. 1.6–51.7] for overweight and obesity, respectively). Despite the association with adverse histopathological features, no association was demonstrated between having overweight or obesity and disease recurrence. A systemic review and meta-analysis by Economides [25] included 11 retrospective cohort studies involving 26,196 participants. The authors reported that BMI was significantly associated with ETE (OR 1.26 [95% C.I. 1.1–1.4] and 1.45 [95% C.I. 1.3–1.6] for overweight and obesity, respectively), multifocality (OR 1.17 [95% C.I. 1.1–1.2] and 1.45 [95% C.I. 1.3–1.6] for overweight and obesity, respectively), increased tumor size (OR 1.77 [95% C.I. 1.5–2] for obesity), and LN metastasis (OR 1.28 [95% C.I. 1.1–1.4] for obesity). However, Matrone and his colleagues [24] found in a well-designed study that included more than 1000 consecutive patients no association between elevated BMI and histopathological and clinical features related to DTC aggressiveness.

Since insulin resistance and adipocytokines that are secreted from unhealthy adipose tissue are components of a mechanism that connects obesity with cancer risk in general [7,35], it would be more reasonable to evaluate the association between obesity and DTC aggressiveness with parameters that better reflect insulin resistance as compared with BMI [27]. A putative support to this notion comes from a recent review entitled “Thyroid cancer and insulin resistance” [36] which emphasizes the importance of insulin resistance in the pathogenesis of DTC, an association which was found in all eight studies that investigated insulin resistance and the metabolic syndrome in this context. Despite our relatively small cohort, we report a positive association between TG/HDL-C ratio and aggressive histopathological variants of DTC (tall cell, columnar, hobnail, and insular variants of PTC together with moderately and poorly differentiated TC). Other histopathological indicators of aggressive disease in patients with overweight and obesity such as ETE and LN metastasis might be explained by late diagnosis, owing to the fact that patients with overweight and obesity typically have thicker necks, which can delay the presentation of a thyroid nodule. However, the presence of aggressive histological variants cannot be attributed to late diagnosis and may be at least partially ascribed to mechanisms stemming from being overweight or obese. To the best of our knowledge, the association between insulin resistance, indicated by worse metabolic parameters (being TG/HDL-C or others), and aggressive histological variants has not been reported before and demands further research.

Our study has noteworthy limitations and strengths. The retrospective nature of the study, together with the relatively small cohort, was a major limitation. In addition, the low absolute number of patients who were presented with adverse histopathological characteristics meant that our sub-group analyses were underpowered. However, patients with DTC in our cohort were well characterized, and we could retrieve high-quality data from the institutional and community electronic databases, which allowed us to analyze the parameters related to metabolic syndrome (e.g., elevated TG and reduced HDL-C levels) which are rarely presented in comparable studies. The studied population was Caucasian (Jewish and Arabs, mostly Bedouins); thus, the generalizability of our results is limited. Further research is warranted to validate our findings across diverse populations.

## 5. Conclusions

In conclusion, our results support previous studies which demonstrated an association between obesity and DTC aggressiveness, as we demonstrated an association between higher BMI and LN metastases. We suggest a link between insulin resistance indices (e.g., higher TG/HDL-C ratio) and aggressive histological variants. We suggest that future studies addressing this association should rely on metabolic parameters which better reflect insulin resistance as compared with BMI.

## Figures and Tables

**Figure 1 healthcare-12-00581-f001:**
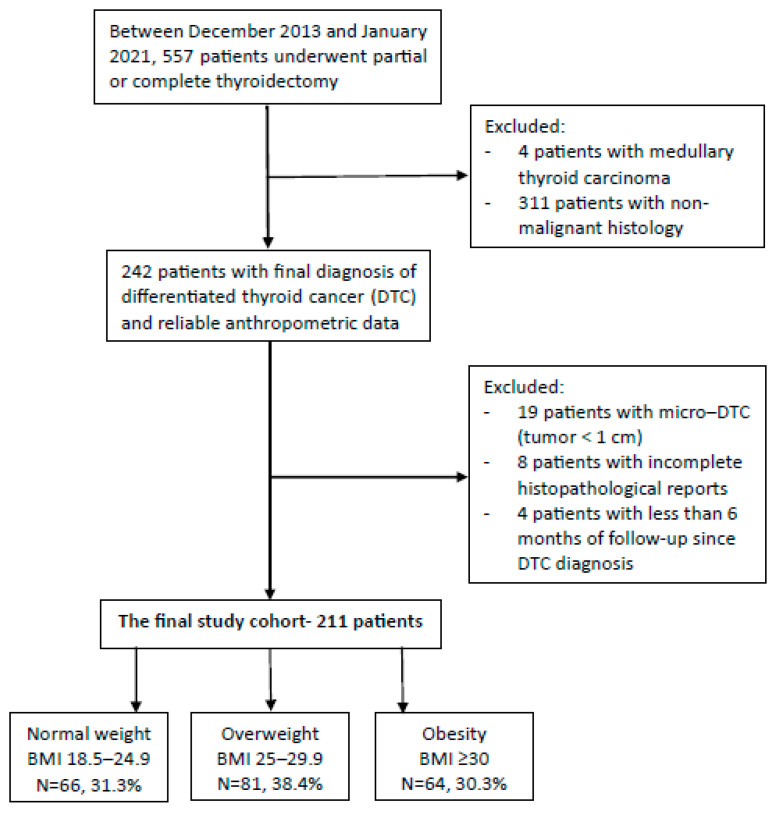
Study flowchart.

**Table 1 healthcare-12-00581-t001:** Baseline characteristics of patients with DTC diagnosis according to their BMI group.

		Normal Weight	Overweight	Obesity	*p* Value *
		BMI 18.5–24.9	BMI 25–29.9	BMI ≥ 30	
		N = 66, 31.3%	N = 81, 38.4%	N = 64, 30.3%	
Age at diagnosis, years (mean ± SD)		41.53 ± 15.37	52.64 ± 16.26	51.14 ± 13.5	* <0.001
					a 0.001
					b <0.001
					c 1
Sex	Females (*n*, %)	57, 86.36%	61, 75.31%	50, 78.13%	* 0.239
					a 0.094
	Males (*n*, %)	9, 13.64%	20, 24.69%	14, 21.88%	b 0.218
					c 0.691
Ethnicity	Jewish (*n*, %)	41, 62.12%	58, 71.6%	35, 54.69%	* 0.106
					a 0.223
	Arab (*n*, %)	25, 37.88%	23, 28.4%	29, 45.31%	b 0.39
					c 0.035
Hypertension (*n*, %)		4, 6.06%	30, 37.04%	19, 29.69%	* <0.001
					a <0.001
					b <0.001
					c 0.433
Pre-diabetes or diabetes mellitus (*n*, %)		6, 9.09%	11, 13.58%	17, 26.56%	* 0.014
					a 0.385
					b 0.007
					c 0.041
Fasting glucose, mmol/L [mg/dL] (median, Q1–Q3)		5.05, 4.77–5.47 [91, 86–98.5]	5.27, 5–5.66 [95, 90–102]	5.49, 5.05–6.22 [99, 91–112]	* 0.001
					a 0.131
					b <0.001
					c 0.111
Glycated hemoglobin (HbA1c), mmol/mol [gr%] (median, Q1–Q3)		37, 31–40 [5.5, 5–5.8]	38, 33–41 [5.6, 5.2–5.9]	40, 37–49 [5.8, 5.5–6.55]	* 0.012
					a 0.925
					b 0.015
					c 0.098
Total cholesterol, mmol/L [mg/dL] (median, Q1–Q3)		4.27, 3.73–5.08 [165, 144–196]	4.64, 4.12–5.34 [179, 159–206]	4.55, 4.04–5.27 [175.5, 156–203.5]	* 0.075
					a 0.093
					b 0.24
					c 1
Triglycerides, mmol/L [mg/dL] (median, Q1–Q3)		1.07, 0.81–1.71 [95, 72–151]	1.33, 1.01–1.81 [118, 89–160]	1.39, 1.06–2.43 [123, 93.5–215]	* 0.014
					a 0.156
					b 0.012
					c 0.801
HDL-C, mmol/L [mg/dL] (mean ± SD)		1.3 ± 0.3 [50.24 ± 11.68]	1.27 ± 0.26 [49.06 ± 10.18]	1.15 ± 0.28 [44.45 ± 10.67]	* 0.009
					a 1
					b 0.012
					c 0.042
Triglycerides/HDL-C (median, Q1–Q3)		2, 1.35–3.34	2.39, 1.72–3.63	2.79, 2.06–5.56	* 0.006
					a 0.395
					b 0.004
					c 0.18
TSH ^¥^, µIU/mL (median, Q1–Q3)		0.45, 0.27–0.7	0.41, 0.28–0.78	0.42, 0.31–0.6	* 0.977

* Comparison of all 3 sub-groups, a—overweight versus normal weight, b—obese versus normal weight, c—obese versus overweight; ^¥^ last result prior to thyroid surgery. Abbreviations: DTC—differentiated thyroid carcinoma, BMI—body mass index (body weight [Kg] divided by the square of height [meters]), N—number, SD—standard deviation, HDL-C—high-density lipoprotein–cholesterol, TSH—thyroid-stimulating hormone.

**Table 2 healthcare-12-00581-t002:** Histopathological features of patients with DTC, according to their BMI group.

			Normal Weight	Overweight	Obesity	*p* Value *
			BMI 18.5–24.9	BMI 25–29.9	BMI ≥ 30	
			N = 66, 31.3%	N = 81, 38.4%	N = 64, 30.3%	
T	pT (*n*, %)	pT 1–2	55, 83.33%	63, 77.78%	50, 78.13%	* 0.763
						a 0.484
		pT 3–4	11, 16.67%	17, 20.99%	13, 20.31%	b 0.563
						c 0.929
	Histological sub-type (*n*, %)	PTC	60, 90.91%	75, 92.59%	60, 93.75%	* 1
						a 1
		FTC	5, 7.58%	6, 7.41%	4, 6.25%	b 1
						c 1
	PTC variant (*n*, %)	Classic	32, 48.48%	25, 30.86%	20, 31.25%	* 0.62
						a 0.6
		Follicular	24, 36.36%	41, 50.62%	30, 46.88%	b 0.332
		Aggressive ^¥^	4, 6.06%	9, 11.11%	10, 15.63%	c 0.69
	Multifocality (*n*, %)		26, 39.39%	38, 46.91%	24, 37.5%	* 0.486
						a 0.403
						b 0.771
						c 0.255
	Any extrathyroidal extension (*n*, %)		14, 21.21%	23, 28.4%	20, 31.25%	* 0.399
						a 0.345
						b 0.232
						c 0.717
LN	LN MTS (*n*, %)	No	52, 78.79%	59, 72.84%	41, 64.06%	* 0.349
						a 0.372
		N1a	6, 9.09%	14, 17.28%	13, 20.31%	b 0.129
		N1b	7, 10.61%	8, 9.88%	9, 14.06%	c 0.576
	Bilateral lateral neck LN MRS (*n*, %)		0, 0%	2, 2.47%	3, 4.69%	* 0.233
						a 0.502
						b 0.116
						c 0.655
	LN diameter ≥ 3 cm ^#^ (*n*, %)		2, 3.03%	0, 0%	4, 6.25%	* 0.037
						a 0.2
						b 0.437
						c 0.036
	Extra-nodal extension ^#^ (*n*, %)		4, 6.06%	7, 8.64%	8, 12.5%	* 0.436
						a 0.754
						b 0.204
						c 0.46
Distal MTS (*n*, %)			4, 6.06%	2, 2.47%	0, 0%	* 0.134
						a 0.407
						b 0.119
						c 0.504

* Comparison of all 3 sub-groups, a—overweight versus normal weight, b—obese versus normal weight, c—obese versus overweight; ^¥^ tall cell, columnar, and insular variants of PTC together with moderately and poorly differentiated variants; ^#^ one or more. Abbreviations: DTC—differentiated thyroid carcinoma, BMI—body mass index (body weight [Kg] divided by the squire of height [meters]), T—primary tumor characteristics, PTC—papillary thyroid carcinoma, FTC—follicular thyroid carcinoma, LN—lymph node/s, MTS—metastasis.

**Table 3 healthcare-12-00581-t003:** Treatment modalities and follow-up data of patients with DTC, according to their BMI group.

		Normal Weight	Overweight	Obesity	*p* Value *
		BMI 18.5–24.9	BMI 25–29.9	BMI ≥ 30	
		N = 66, 31.3%	N = 81, 38.4%	N = 64, 30.3%	
Thyroid surgery (*n*,%)	Partial	23, 34.85%	15, 18.52%	17, 26.56%	* 0.08
					a 0.024
	Complete	43, 65.15%	66, 81.48%	47, 73.44%	b 0.306
					c 0.246
RAI treatment (*n*,%)	No	31, 46.97%	30, 37.04%	29, 45.31%	* 0.839
					a 0.532
	30 mCi	10, 15.15%	16, 19.75%	11, 17.19%	b 0.948
	≥100 mCi	21, 31.82%	27, 33.33%	20, 31.25%	c 0.697
Risk of disease recurrence ^¥^ (*n*,%)	Low	37, 56.06%	36, 44.44%	29, 45.31%	* 0.363
					a 0.228
	Intermediate or high	25, 37.88%	39, 48.15%	30, 46.88%	b 0.276
					c 1
Response to treatment ^§^ (*n*,%)	Excellent	49, 74.24%	57, 70.37%	37, 57.81%	* 0.108
					a 0.79
	Incomplete ^#^	7, 10.61%	7, 8.64%	12, 18.75%	b 0.111
					c 0.056
Additional intervention ^@^ (*n*,%)		0, 0%	5, 6.17%	5, 7.81%	* 0.049
					a 0.062
					b 0.024
					c 0.75

* Comparison of all 3 sub-groups, a—overweight versus normal weight, b—obese versus normal weight, c—obese versus overweight; ^¥^ risk of structural disease recurrence; ^§^ response to treatment as was documented at the last follow-up visit; ^#^ biochemical and/or structural [28]; ^@^ second thyroid operation, and/or neck dissection, and/or second RAI treatment, and/or external beam irradiation. Due to missing data in some variables the total percent per column might not summarize to 100%. Abbreviations: DTC—differentiated thyroid carcinoma, BMI—body mass index (body weight [Kg] divided by the square of height [meters]), RAI—radioactive iodine, mCi—millicurie.

**Table 4 healthcare-12-00581-t004:** Association between BMI as continuous variables and selected histopathological features.

Histopathological Feature	Sub-Categories	Unadjusted	Adjusted ^@^
		Odds Ratio (95% CI) *	Odds Ratio (95% CI) *
Histological sub-type and variants	PTC—classic variant	1.00 (REF)	1.00 (REF)
	PTC—Follicular variant	1.043 (0.982–1.107)	1.024 (0.962–1.09)
	Aggressive variant ^¥^ of PTC and/or FTC	1.089 (1.011–1.172)	1.055 (0.973–1.143)
Extrathyroidal extension	None	1.00 (REF)	1.00 (REF)
	Microscopic	1.016 (0.997–1.129)	1.061 (0.993–1.133)
	Gross	0.968 (0.869–1.079)	0.915 (0.806–1.04)
Vascular invasion	No	1.00 (REF)	1.00 (REF)
	Yes	0.984 (0.92–1.052)	0.984 (0.917–1.056)
Multifocality	No	1.00 (REF)	1.00 (REF)
	Yes	1.004 (0.954–1.057)	0.997 (0.943–1.054)
Any lymph node metstasis	None	1.00 (REF)	1.00 (REF)
	Any	1.061 (1.003–1.123)	1.077 (1.013–1.145)
Lateral neck LN MTS	No	1.00 (REF)	1.00 (REF)
	Yes	1.117 (0.978–1.275)	1.129 (0.969–1.316)
LN MTS ≥ 3 cm	No	1.00 (REF)	1.00 (REF)
	Yes	1.084 (0.952–1.233)	1.101 (0.966–1.255)
Extra-nodal extension	No	1.00 (REF)	1.00 (REF)
	Gross	1.040 (0.958–1.130)	1.037 (0.948–1.135)
Distal MTS	No	1.00 (REF)	1.00 (REF)
	Yes	0.863 (0.709–1.05)	0.81 (0.645–1.017)

^@^ Adjusted for age, sex, and ethnicity; * bolded for *p* ≤ 0.05; ^¥^ tall cell, columnar, and insular variants of PTC together with moderately and poorly differentiated variants. Abbreviations: BMI—body mass index (body weight [Kg] divided by the square of height [meters]), CI—confidence interval, PTC—papillary thyroid carcinoma, FTC—follicular thyroid. carcinoma, LN—lymph node/s, MTS—metastasis.

**Table 5 healthcare-12-00581-t005:** Association between triglycerides/HDL-C as continuous variables and selected histopathological features of patients with DTC aged 30–70 years at the time of the index operation.

Histopathological Feature	Sub-Categories	Unadjusted	Adjusted ^@^
		Odds Ratio (95% CI) *	Odds Ratio (95% CI) *
Histological sub-type and variants	PTC classic variant	1.00 (REF)	1.00 (REF)
	PTC—follicular variant	1.13 (0.942–1.355)	1.125 (0.922–1.372)
	Aggressive variant ^¥^ of PTC and/or FTC	1.302 (1.047–1.619)	1.269 (1.001–1.61)
Extrathyroidal extension	None	1.00 (REF)	1.00 (REF)
	Microscopic	1.007 (0.839–1.208)	0.999 (0.819–1.219)
	Gross	0.839 (0.581–1.211)	0.716 (0.465–1.104)
Vascular invasion	No	1.00 (REF)	1.00 (REF)
	Yes	0.995 (0.805–1.229)	0.999 (0.796–1.255)
Multifocality	No	1.00 (REF)	1.00 (REF)
	Yes	1.137 (0.975–1.326)	1.072 (0.908–1.266)
Lateral neck LN MTS	No	1.00 (REF)	1.00 (REF)
	Yes	1.029 (0.658–1.61)	0.931 (0.572–1.517)
LN MTS ≥ 3 cm	No	1.00 (REF)	1.00 (REF)
	Yes	1.25 (0.893–1.749)	1.163 (0.796–1.699)
Extra-nodal extension	No	1.00 (REF)	1.00 (REF)
	Yes	1.101 (0.845–1.435)	1.022 (0.759–1.376)
Distal MTS	No	1.00 (REF)	1.00 (REF)
	Yes	1.146 (0.652–2.016)	1.134 (0.481–2.675)

^@^ Adjusted for age, sex, and ethnicity; * bolded for *p* ≤ 0.05; ^¥^ tall cell, columnar, and insular variants of PTC together with moderately and poorly differentiated variants. Abbreviations: HDL-C—high-density lipoprotein–cholesterol, CI—confidence interval, PTC—papillary thyroid carcinoma, FTC—follicular thyroid carcinoma, LN—lymph node/s, MTS—metastasis.

## Data Availability

Data are available upon request and according to the national policy of data sharing requiring authorization by the ethics committee.

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
