# Peer review of "Are Higher Body Mass Index and Worse Metabolic Parameters Associated with More Aggressive Differentiated Thyroid Cancer? A Retrospective Cohort Study"

_healthcare, 2024, doi:10.3390/healthcare12050581_

Round 1

Reviewer 1 Report

Comments and Suggestions for Authors

The authors present interesting work on investigating the association between BMI/ other metabolic parameters and thyroid cancer aggressiveness. It is established that high BMI is a risk factor for thyroid cancer but the potential link with more aggressive cancer is currently still understudied. Please find my comments below.

Methods:

-          Line 105: the authors use 18.5 kg/m2 as the cut off for normal weight. Can the authors confirm that no patients were underweight < 18.5 kg/m2?

-          Can the authors add what kind of hospital the patients were recruited from (specialized clinic? Tertiary center?) This may impact the results depending on whether more complex cancers were seen at this particular hospital.

-          Can the authors explain why patients with a tumor size smaller than 1cm were excluded from the study? A large number of patients with DTC have small tumors. The results of this studies would therefore not be generalizable to the DTC population if excluding < 1 cm tumors.

-          Please add the number of patients excluded per group to the paper (methods; line 112-116)

Results:

-          Line 166-168: The authors mention that most DTC patients have low risk histopathological features. Please define in the methods what would be considered low risk features?

-          Would it be possible to add information about minor/ gross ETE to the manuscript? Gross ETE is a better prognostic feature than minor ETE.

-          Line 193 and line 240 (discussion): the authors use the term trend in this sentence. Trend suggests a change over time. I suggest to rephrase and avoid the use of the term trend

-          The term “risk assessment” in table 3 is confusing. Can the authors use “disease recurrence”? If not, please add a description to the methods about “risk assessment”

-          Line 216: results reported are the results of the adjusted analysis. Please specify this in the text.

-          Table 4 and 5: the notation of the reference groups is confusing. Proper notation is to include the group in the column of sub-categories and then add 1.00 (REF) to the columns of results.

Discussion:

-          The authors repeat their results in the discussion (e.g. line 254-255). I suggest to remove the result from the discussion.

-          The explanation of choosing 30 and 70 years as upper and lower limits for the subgroup analyses (line 258-262) should be moved to the methods.

-          The authors resummarize the results from previous studies now adding in OR without 95% CI. I suggest to add the 95%CI. Moreover, this feels like a repetition of the introduction. The discussion could be improved with more in depth information about the potential link (e.g. in vitro/ animal studies on pathways?)

Comments on the Quality of English Language

Minor comments:

-          Please correct typo in table 2: “any eetrathyroidal extension”

-          Please correct typo in line 189-190: “patrial”

Reviewer 2 Report

Comments and Suggestions for Authors

The authors evaluated the association of BMI and other metabolic parameters with differentiated thyroid cancer aggressiveness in a surgical cohort. Overall the manuscript is well-written and this is an interesting study. I have a few comments that I think could help strengthen the presentation of the methods and results.

  • In line 138, please state if the p-value was two sided or one sided.
  • In lines 130-132, please provide the statistical tests that you used to compare between three groups. Student’s t test and Mann-Whiteney test can only be used for comparing between two groups. Did you use the ANOVA test or the Kruskal-Wallis test for comparing more than two groups? Or did you perform multiple comparisons between each pair among the three groups? If you conducted multiple comparisons, please describe how you corrected the p-values for multiple comparisons.
  • Lines 135-136, what do you mean by multinomial logistic regression analysis as well as logistic regression? Instead of saying logistic regression, I think you mean to say binary logistic regression or dichotomous logistic regression.
  • For the study population, could you provide the number of people who were excluded due to each exclusion criterion? A population flow chart could help illustrate that. What are the reasons that some patients have less than 6 months of follow-up data following histopathological diagnosis? Is it because they were diagnosed more recently?
  • Please also describe the location of your institution. It could help readers understand your study target population and the generalizability of your results.
